# Advances and Hurdles in CAR T Cell Immune Therapy for Solid Tumors

**DOI:** 10.3390/cancers14205108

**Published:** 2022-10-18

**Authors:** Francesco Boccalatte, Roberto Mina, Andrea Aroldi, Sarah Leone, Carter M. Suryadevara, Dimitris G. Placantonakis, Benedetto Bruno

**Affiliations:** 1Department of Pathology, NYU Grossman School of Medicine, New York, NY 10016, USA; 2Perlmutter Cancer Center, NYU Langone Health, New York, NY 10016, USA; 3Division of Hematology, A.O.U. Città della Salute e della Scienza di Torino, University of Torino, 10126 Torino, TO, Italy; 4Department of Molecular Biotechnology and Health Sciences, University of Torino, 10126 Torino, TO, Italy; 5Department of Medicine and Surgery, University of Milano-Bicocca, 20900 Monza, MB, Italy; 6Department of Population Health, NYU Grossman School of Medicine, New York, NY 10016, USA; 7Department of Neurosurgery, NYU Grossman School of Medicine, New York, NY 10016, USA; 8Brain and Spine Tumor Center/Neuroscience Institute, NYU Grossman School of Medicine, New York, NY 10016, USA

**Keywords:** solid tumors, chimeric antigen receptor (CAR) T cell, adoptive immunotherapy, receptors, chimeric antigen, tumor microenvironment, xenograft models

## Abstract

**Simple Summary:**

Chimeric antigen receptor (CAR) T cells are genetically engineered T cells that recognize markers present on tumor cells and drive the degradation of the tumor itself. CAR T immunotherapy has obtained remarkable success in targeting a number of blood malignancies; however, its outcome is typically modest when applied to solid tumors, because of specific structural, biological, and metabolic aspects of the solid tumor environment. This article offers an overview of the interactions between CAR T cells and the solid tumor microenvironment, highlighting the main strategies that have been attempted to overcome CAR T suppression, both in preclinical models and in clinical trials.

**Abstract:**

Chimeric antigen receptor (CAR) T cells in solid tumors have so far yielded limited results, in terms of therapeutic effects, as compared to the dramatic results observed for hematological malignancies. Many factors involve both the tumor cells and the microenvironment. The lack of specific target antigens and severe, potentially fatal, toxicities caused by on-target off-tumor toxicities constitute major hurdles. Furthermore, the tumor microenvironment is usually characterized by chronic inflammation, the presence of immunosuppressive molecules, and immune cells that can reduce CAR T cell efficacy and facilitate antigen escape. Nonetheless, solid tumors are under investigation as possible targets despite their complexity, which represents a significant challenge. In preclinical mouse models, CAR T cells are able to efficiently recognize and kill several tumor xenografts. Overall, in the next few years, there will be intensive research into optimizing novel cell therapies to improve their effector functions and keep untoward effects in check. In this review, we provide an update on the state-of-the-art CAR T cell therapies in solid tumors, focusing on the preclinical studies and preliminary clinical findings aimed at developing optimal strategies to reduce toxicity and improve efficacy.

## 1. Introduction

Despite the dramatic expansion of chimeric antigen receptor (CAR) T cell therapies in hematological malignancies since their introduction in 2012, their application in solid tumors has faced several hurdles owing to antigen heterogeneity, suboptimal CAR T cell trafficking, and the numerous immunosuppressive features of the tumor microenvironment (TME), which causes T cell dysfunction and greatly inhibits the effectiveness of CAR T cells. In this review we describe the most important preclinical and preliminary clinical findings regarding CAR T cells against solid tumors.

## 2. The Solid Tumor Microenvironment and Its Impact on CAR T Therapy

The tumor microenvironment (TME) is a complex ecosystem where tumor cells interact with soluble factors (cytokines, chemokines), immune cells (e.g., lymphocytes, phagocytic cells, and antigen-presenting cells), and non-immune cells (such as endothelial and stromal cells). Moreover, physical, metabolic, and biochemical factors contribute to the TME, and have a significant impact on the natural immune responses and on immunotherapies (Figure 1). Though solid tumors represent the majority of cancers, adoptive immunotherapies, including the most innovative CAR T cells, have been far less efficient in this setting as compared to hematological malignancies. In the next paragraphs, we illustrate factors that impair the potential efficacy of CAR T therapy and highlight possible strategies to overcome these hurdles.

### 2.1. CAR T Cell Trafficking and Infiltration

In the setting of solid tumors, immune cells should efficiently reach tumor sites and infiltrate masses that can be of considerable size. Successful immune cell trafficking depends on the concordant expression of chemokines secreted by the tumor, and the appropriate chemokine receptors on the T cells. Similarly, the infiltration process is driven by a matched expression of adhesion receptors/ligands by the T cells and the tumor endothelium. Unfortunately, tumors often downmodulate the expression of chemoattractant molecules, therefore escaping immune surveillance [1]. Recent studies have profiled the cytokine/chemokine profile of several tumors to better identify targets for CAR T. For instance, given that CD70+ gliomas produce high levels of interleukin IL-8, Jin et al. generated anti-CD70 CAR T cells that express high levels of IL-8R, demonstrating a more efficient recruitment to the tumor site than CD70-CAR alone [2]. Similarly, lung tumors secreting high levels of CCL2 can be efficiently eradicated by CAR T cells targeting mesothelin and concomitantly expressing the CCL2 receptor CCR2b [3,4]. Another strategy is based on oncolytic viruses, which specifically infect tumor cells. These viruses can be engineered to express chemotactic chemokines, which, in turn, attract CAR T cells to the tumor site [5]. Local instillation is another tool to facilitate CAR T cell trafficking. It was initially believed to be efficient only on local injection sites while sparing metastasis. However, recent studies have demonstrated that it is also possible to obtain tumor eradication at distant sites [6,7]. Recently, the field of locoregional administration of CAR T cells further expanded towards creating scaffolds that facilitate the expansion, permanence, and release of CAR T cells on site. This can be successfully achieved by biopolymer scaffolds, often armored with stimulatory molecules to activate the APC T cell axis [8], or by nitinol films that can host a larger cargo of CAR T cells to be implanted on tumor sites [9].

### 2.2. The Solid Tumor Microenvironment: Physical and Metabolic Barriers

Several physical barriers hamper the accessibility of CAR T cells to a solid tumor mass, including thick surrounding tumor stroma, aberrant vasculature, and high interstitial pressure. The stroma is mostly constituted by cancer-associated fibroblasts (CAFs) that drive the deposition of the extracellular matrix (ECM), thus physically preventing the infiltration of immune cells. Several strategies are under investigation to overcome this obstacle. CAR T cells can be engineered to target fibroblast activation protein (FAP), therefore reducing the number of CAFs in the microenvironment [10]. Alternatively, CAR T cells can be armored with proteases that degrade the ECM. For instance, CAR T cells engineered to express heparanase, which degrades the heparan sulfate component of the ECM, have shown better infiltration and tumor clearance both in vitro and in animal models [11]. Moreover, the aberrant tumor vasculature causes interstitial hypertension that prevents extravasation and a hypoxic microenvironment, especially in the central part of the tumor. Thus, normalizing the tumor vasculature may be beneficial [12]. In this context, vascular endothelial growth factor (VEGF) signaling plays a pivotal role. Antiangiogenic therapy that blocks VEGF signaling improves immune cell infiltration [13], and anti-VEGFR CAR T cells can efficiently inhibit tumor growth—as shown in several syngeneic mouse models [14].

In terms of metabolic barriers, it is worth noticing that the particular anatomical structure of solid tumors generates hostile hypoxia and nutrient starvation for immune cells. The hypoxic environment caused by poor perfusion and abnormal vasculature hampers the expansion of CAR T cells and shifts their phenotype from effector to central memory [15]. Strategies to favor CAR T response in the hypoxic TME are under investigation by fusing an oxygen-sensitive domain of hypoxia-inducible factor 1 (HIF1a) to the CAR scaffold [16].

Of note, to mount an effective antitumor response, CAR T cells need to proliferate and produce cytokines and molecules that degrade tumor cells. Thus, CAR T cells must compete for nutrients and metabolites in a niche where tumor cells are scavenging most resources. CAR T cell effector functions rely on glucose and glycolytic metabolism [17], which becomes highly challenging in a nutrient-poor environment. In particular, it has been observed that the insufficient production of phosphoenolpyruvate (PEP) in T cells can dampen TCR signaling, and therefore limit the effector response, and that PEP supplementation can efficiently restore T cell responses [18]. It should also be noted that the addition of specific costimulatory molecules to the CAR structure has an impact on their glycolytic or fatty acid metabolism, and therefore their effectiveness in combating tumor cells [19]. Moreover, the lack of amino acids, such as tryptophane, lysine, and arginine, activates the stress response in T cells, causing the shutdown of protein translation and autophagy response [20,21]. Tumor cells also release into the TME a series of molecules, such as kynurenine, adenosine, and lactate, that inhibit T cells. Lactate contributes to lower the pH of the TME, where elevated acidity impairs T cell proliferation and the production of cytokines and lytic enzymes, such as perforin and granzyme B. Counterbalancing the acidity of the TME using bicarbonate or proton pump inhibitors can partially reverse T cell anergy [22]. Recent studies have observed that competition for nutrients and metabolic imbalances in the TME may also drive T cell mitochondrial dysfunction and cause the exhaustion and depletion of both endogenous T and CAR T cells [23]. Overall, immunometabolism is an emerging field in cancer research, and many seminal studies are proving that the modulation of metabolic pathways within the TME is a promising avenue to improve the efficacy of immunotherapy.

### 2.3. The Solid Tumor Microenvironment: Soluble and Cellular Drivers of Immune Suppression

The TME is replete with soluble factors released by both tumor and immune cells. Most of them have direct suppressive roles in CAR T-cell-driven adoptive immunotherapy. Most of these factors have a direct suppressive role on CAR T cell adoptive immunotherapy. For example, adenosine is an immunosuppressive metabolite secreted by tumor and immune cells in the TME. In melanoma models, antagonists of the adenosine 2a receptor strongly increase the efficacy of CAR T therapy, either alone or in combination with a PD-1 checkpoint blockade [24]. Another inhibitory factor is prostaglandin E2 (PGE2), an inflammatory molecule generated by tumor cells and macrophages that impairs CD4+ T cell proliferation and CD8+ T cell differentiation [25]. Both PGE2 and adenosine exert their immunosuppressive function by activating protein kinase A (PKA), which inhibits TCR signaling. Newick et al. engineered CAR T cells to express a PKA inhibitor peptide, and showed that these armored CAR T cells had improved TCR signaling, cytokine production, and enhanced tumor killing [26].

An important class of soluble factors in the TME is represented by cytokines and chemokines. These molecules can function either as boosters or inhibitors of antitumor responses. In the solid TME, cytokines act not only by impairing cytotoxic T cells, but also by recruiting immunosuppressor cells from peripheral sites, and by polarizing the resident immune cells towards an immunosuppressive phenotype. The most widely studied inhibitory cytokine in the context of the TME is tumor growth factor beta (TGFb). This factor acts both on the tumor stroma, where it enhances matrix deposition and shields tumor cells from immune surveillance [27], and on T cells, where it inhibits effector functions and skews their phenotype towards immune tolerance [28]. The systemic blockade of TGFb receptor signaling has been shown to enhance the efficacy of adoptive immunotherapy [29]. Other studies have generated synthetic receptors to target TGFb signaling, such as a TGFb dominant negative (DN) receptor and a TGFb CAR. The TGFb DN receptor is a truncated, non-functional form of TGFb receptor that cannot transduce the intracellular signal, and therefore competes with the natural TGFb ligand–receptor function [30]. The TGFb CAR has a double function, since it outcompetes the natural TGFb receptor for binding its ligand, and additionally stimulates the antitumor activity of neighboring cytotoxic T cells [31].

Other inhibitory cytokines belong to the family of interleukins, for example IL-10 and IL-4. To counteract the inhibitory effect of IL-4, two different groups have engineered chimeric IL-4 receptors by fusing its extracellular domain with either the intracellular domain of the IL-2 receptor [32] or the intracellular domain of the IL-7 receptor [33]. These chimeric receptors can be combined with other approaches, and have been efficiently used to boost adoptive immunotherapy in animal models [34]. Further studies have tried to increase the release of inflammatory cytokines, such as IL-12, in the TME to favor adoptive immunotherapy. CAR T cells that release IL-12 upon their activation can boost the natural immune cell response towards tumor cells that are escaping immunotherapy [35]. Although this approach has achieved promising results in animal models, the high toxicity of IL-12 has so far hampered its clinical application. New classes of CAR T cells “armored” with less toxic proinflammatory cytokines are currently under study [36].

Together with soluble factors, many different cell types harbor in the solid TME. Of note, suppressive cell populations are found both in the myeloid and in the lymphoid lineage. Regulatory T cells (Tregs), myeloid-derived suppressor cells (MDSCs), tumor-associated macrophages (TAMs), and tumor-associated neutrophils (TANs) have been extensively studied. Some of these cells, for instance TAMs, derive from an intrinsic proinflammatory and antitumorigenic macrophage phenotype, the so called M1 phenotype, characterized by Th1 cytokine secretion, but in the solid TME they convert into an anti-inflammatory and protumorigenic phenotype, the M2 phenotype, characterized by Th2 cytokine secretion [37,38].

The inhibitory effect of myeloid immunosuppressive cell populations on CAR T cells is currently a prominent area of research. As previously described, most immunosuppressive cells (TAMs, TANs, MDSCs) constantly release soluble factors, such as TGFb, PGE2, and IL-10, impairing CAR T cell functions [38]. Moreover, myeloid-derived suppressive cells express on their surface the programmed cell death ligand 1 (PDL-1), which acts as an inhibitory stimulus while binding to the PD1 receptor on T cells [39]. Based on this observation, several studies have shown that PD1 blockade improves the therapeutic efficacy of CAR T cells on solid tumors [40]. Other groups have instead focused on depleting or re-educating the suppressor cell types. For example, the inhibition of colony-stimulating factor 1 receptor (CSF1R) can selectively deplete TAMs from the TME, which results in increased tumor killing by resident T cells [41]. Similarly, blockade of the macrophage receptor with collagenous domain (MARCO) can re-educate TAMs from an immunosuppressive phenotype to an immunoactivating phenotype, thus enhancing tumor killing [42].

In the lymphoid lineage, CD4+/FOXP3+ T regs are known to inhibit T cell activity at multiple stages, either via the secretion of suppressive factors (TGFb, IL-10, IL-35, adenosine), by cell-to-cell contact, or through competition for activating cytokines [43]. Given their prominent role in generating immune tolerance and impairing T cell functions, several approaches have been attempted to deplete Tregs in the TME. However, since most Treg markers are shared with other cell populations, including CAR T cells, selective depletion of Tregs while sparing antitumor efficacy remains highly challenging [44]. Recent evidence has shown that lymphodepletion prior to CAR T cell infusion has an important role in allowing durable responses [45]. Finally, different types of stromal cells may affect the antitumor response. Cancer-associated fibroblasts (CAFs) are significantly distinct from normal fibroblasts, and play different roles in shielding cancer cells from adoptive immunotherapy. CAFs both secrete an abundant extracellular matrix and several growth factors that support tumorigenesis, including VEGF, which remodels tumor vasculature [46]. Very recently, CAF targeting was shown to improve CAR T cell therapy in myeloma [47], but a similar approach has not been investigated in solid tumors yet.

## 3. In Vivo Models to Study CAR T Cell Therapy for Solid Tumors

In vivo mouse models have always been considered a decisive method to validate in vitro results in cancer immunotherapy and chimeric antigen receptor (CAR) T cell therapy, owing to the fact that mice are low-cost animals with a short reproductive cycle, where tumor cells can easily grow at high proliferation rate [48]. Nonetheless, the choice of the right model becomes crucial to reproduce all the corresponding conditions that reflect the characteristics of tumors in the clinical setting. Reproducibility of real conditions should be the main goal while setting in vivo experiments, and the identification of proper mouse strains is the main critical point to consider.

### 3.1. Syngeneic Mouse Models

The immunocompetent mouse model (i.e., C57BL/6, BALB/c, and FVB) harboring murine cancer cell lines (i.e., the syngeneic mouse model) is the oldest and most standardized approach to study anticancer treatment, providing fast tumor growth after transplantation of murine cancer cells [49]. In this immunocompetent setting, CAR T cells—together with tumors and target antigens—have murine origins, which means that mouse T cells are collected and infected with a lentiviral/retroviral vector to express the CAR construct of interest before in vivo injection [50]. This system provides important information about efficacy and on-target off-tumor toxicities, as healthy mouse tissues might potentially express low levels of target antigens, similarly reflecting antigen pattern conditions of human patients [51]. As a matter of fact, anti-CD19 CAR T cells in an immunocompetent C57BL/6 mouse model showed successful efficacy in eradicating the CD19+ murine Eμ-ALL01 cell line with simultaneous onset of B cell aplasia as an on-target off-tumor effect, which resembles the results and safety profile obtained in clinical practice [52,53]. Conversely to hematological models, preclinical syngeneic mouse models of CAR T cell therapy in solid tumors has turned out to be more challenging, owing to the fact that murine solid cancer cell lines harbor fast proliferation rates immediately after in vivo injection: this leads to rapid tumor growth without simultaneous development of proper inflammatory environment, which characterizes the hostile TME in solid cancer [54]. The lack of proper TME in preclinical models prevents the possibility of studying any potential interactions between CAR T cells and TME, thus explaining possible unsuccessful results in some clinical settings [55,56]. To improve the probability of in vivo TME formation, orthotopic injection of mouse cancer cell lines in the corresponding organ might be an option [57]. Indeed, this approach could better reflect the TME if compared to subcutaneous injection, even though the procedure is much more complicated and difficult to reproduce owing to the fact that special expertise and proper equipment are required for appropriate tumor implantation and monitoring (i.e., intrapancreatic injection of pancreatic duct adenocarcinoma, intracranial injection of glioblastoma cell lines) [57]. In syngeneic mouse models of solid cancer, the possibility of studying the interaction between TME and CAR T cells has contributed towards clarifying the pathogenetic aspects underlying impaired CAR T activity within hostile TME. In the case of liver metastases of mouse epithelial colon carcinoma expressing carcinoembryonic antigen (CEA), it has been shown that tumor cells are sensitive to anti-CEA CAR T cells, whose efficacy is in turn influenced by high intrahepatic levels of myeloid-derived suppressive cells (MDSCs) belonging to hostile TME, and able to dramatically quench CAR T activity [58]. As a result, syngeneic mouse models can provide critical information in terms of CAR T efficacy and on-target off-tumor effects. However, the reproducibility of human conditions remains a concern, as well as the lack of tumor heterogeneity. Mouse-derived cultured cancer cell lines are genetically uniform, in contrast with the natural genomic complexity of cancers. Moreover, the possibility of creating proper conditions of cytokine release syndrome (CRS) and neurotoxicity (NE) in these models is limited [50,59]. Finally, further limitations of syngeneic mouse models are located in the murine CAR construct design, which might only have affinity for mouse antigens, thus requiring backbone modifications to obtain a CAR construct with proper affinity against the corresponding human antigen [50]. Therefore, different mouse and human CAR constructs might not reflect all the possible on-target and off-tumor toxicities, raising concerns in terms of safety for subsequent clinical applications [50,59].

### 3.2. Human Xenograft Mouse Models

To evaluate CAR T cells as an anticancer treatment, human tumors cannot be studied in immunocompetent mice given that the murine immune system can easily eradicate human tumor cells [55]. Immunodeficient mouse models have been developed over the last few decades to improve the efficiency of human tumor engraftment [60]. The athymic Foxn1*^nu^* (nude) mouse was the first xenograft model available, based on abnormal development of the thymus, which exclusively defines dysfunctional T cells. However, residual innate immune components still have an impact in preventing human tumor engraftment (i.e., neutrophils, dendritic cells, natural killer (NK) cells, and B cells) [61]. To further reduce tumor rejection, non-obese diabetic/severe combined immunodeficient (NOD/SCID) mice have been developed, harboring only residual NK cell activity, which provides better human tumor engraftment than nude mice [62]. However, many cancers and hematological malignancies fail to properly engraft owing to the residual NK cell activity that confers tumor rejection [63]. Finally, immunodeficient NOD/SCID mice bearing a targeted mutation in the interleukin-2 (IL-2) receptor common gamma chain gene (IL2rγ^null^) were developed. This gene modification defined the NOD/SCID/IL2Rγ^null^ (NSG) strain, which became the most common mouse model used to study human cancer and CAR T cell activity [50]. The IL2rγ gene is responsible for high-affinity signaling for the IL-2, IL-4, IL-7, IL-9, IL-15, and IL-21 receptors, which compromises both the adoptive and the innate immune system [64,65]. Hence, this immunodeficient system turned out to be the most receptive for human-derived cultured cancer cell lines and primary tumors to date [60].

### 3.3. Xenograft Models and CAR T Cells

Human-derived cultured tumor cell lines can easily grow in NSG mouse models, and, after CAR T cell administration, the tumor burden can be monitored either by direct size measurement—in case of subcutaneous injection—or by indirect monitoring after orthotopic or tail vein injection [57,58,66,67,68,69,70]. In this setting, however, on-target off-tumor toxicity can be assessed only if the CAR construct harbors affinity both for tumor-associated antigen/s and its/their corresponding murine ortholog in mouse tissues. Unfortunately, this is not frequent. One study employed CAR T cells with multiple affinities against human and mouse ErbB dimer receptors, providing information in terms of safety and on-target off-tumor activity in an immunocompromised xenograft mouse model [70]. Human transgenic xenograft models expressing human antigens on mouse tissues are being developed to explore on-target off-tumor toxicity when CAR T constructs only recognize human-derived antigens. A recent NSG mouse model, expressing the human epidermal growth factor receptor 2 (HER2) molecule on liver tissue after adenoviral or transposon gene delivery, turned out to be a valid system to detect undesired toxicity. It also showed that greater liver damage occurred when high-affinity CAR T was used, whereas less damage occurred in the case of low-affinity CAR T. Of note, tumor control was preserved [71]. Overall, the NSG mouse system still presents technical downsides due to the lack of tumor heterogeneity as well as the presence of non-physiological interactions between mouse TME and human-derived CAR T cells [59].

To address these issues, transplantation of primary patient tumors in NSG mouse models (patient-derived xenograft models) became an accurate option that better reflected tumor complexity by mirroring in vivo natural tumor behavior [72]. Unlike tumor cell lines, primary tumors are not affected by genomic alterations that occur during in vitro multiple passages, preserving primary tumor genetics and showing real-world disease conditions [60]. For these reasons, patient-derived xenograft (PDX) models have recently been employed in the preclinical development of CAR T cell therapy [50], showing promising results [73,74,75]. Subcutaneous implanted PDX samples from colorectal cancer were efficiently treated by anti-HER2 human CAR T cells [74]. Other PDX models included anti-B7-H3 CAR T cells for pancreatic cancer and anti-EGFRvIII CAR T for glioblastoma [73,75]. PDX tumor monitoring can be performed either by direct subcutaneous evaluation or by non-invasive bioluminescence imaging (BLI) [66,67]. Overall, TME modeling remains suboptimal in NSG mice, which lack an immune system capable of interacting with human tumors [60]. To overcome this issue, “humanized xenograft models” were created by obtaining the engraftment of hematopoietic stem cells (HSCs) in NSG mouse models and allowing for a robust immune reconstitution [60]. NSG mice can be injected with HSCs that differentiate into a complex human immune system with the emergence of helper and cytotoxic T cells, B cells, monocytes, dendritic cells, and NK cells [76]. The risk of graft versus host reaction (GvHR) is minimal since the thymic selection of human T cells occurs in the context of mouse major histocompatibility complex (MHC) molecules [77,78]. Furthermore, both autologous and allogeneic HSCs can be used in the humanized setting [79,80,81,82]. Humanized models for CAR T cell therapy have been developed especially to provide more reliable data on CAR T cell toxicity. B cell aplasia, seen in the clinical setting, was confirmed in humanized NSG mice harboring PDX B-ALL treated with anti-CD19 CAR T cells. Likewise, transient monocytopenia was observed when anti-CD44v6 CAR T cells were employed to treat humanized models of acute myeloid leukemia and multiple myeloma [83,84]. Furthermore, humanized models finally provided consistent findings on CRS and NE identifying the monocyte–macrophage system as the main source of cytokine production (e.g., IL-1, IL-6), responsible for human CAR T expansion and the accompanying onset of CRS and NE [85,86]. Moreover, additional observations showed that anti-IL-6 signaling blockade was useful to control CRS rather than NE, providing critical insights for the clinical management of CAR T-cell-associated toxicity [85,86,87].

In the setting of solid tumors, the establishment of the human immune system in NSG mice greatly helps in assessing its interaction with adoptive T cell therapies, thus leading to a better understanding of the crosstalk between the TME and the CAR T axis [59]. One study showed that renal cell carcinoma and NK-cell-transplanted NSG mice were efficiently treated with carbonic anhydrase IX (CAIX)-targeted CAR T cells, which also secreted the anti-PD-L1 antibody. The secreted antibody, by binding to the Fc receptor on the NK cell surface, was able to recruit human NK cells to the tumor site, which further improved in vivo tumor killing [88]. Another humanized model confirmed the detrimental effects of myeloid-derived stem cells (MDSCs) in TME on CAR T cell activity, showing that a consistent reduction in intratumoral MDSCs provided a better CAR T response against a xenograft model of neuroblastoma [89]. This study employed a combination of NKG2D.ζ-NK cells (NK cells with an NKG2D receptor fused to the cytotoxic ζ-chain of the T cell receptor) and anti-GD2 CAR T cells, owing to the fact that GD2 is overexpressed in neuroblastoma [90]. Since NKG2D ligands were in turn expressed at high levels by MDSCs, NKG2D.ζ-NK cells were able to eliminate the suppressive myeloid cells in the tumor, allowing anti-GD2 CAR T cells to persist and be active within the TME [89,91]. In this elegant study, a xenograft model of neuroblastoma and reconstituted TME by subcutaneous co-injection into NSG mice of a human neuroblastoma cell line and human MDSCs was designed [89]. These mice were then treated with NKG2D.ζ-NK cells, followed by anti-GD2 CAR T cells infusion. Within the tumor, MDSCs were completely eliminated by the modified NK cells in vivo: this was associated with an increase in intratumoral proinflammatory cytokines, which led to higher recruitment and infiltration of CAR T cells within TME, providing stronger tumor regression in NKG2D.ζ-NK/anti-GD2 CAR T-cell-treated mice with respect to the control group only treated with anti-GD2 CAR T cells [89]. In summary, the introduction of humanized mouse models allowed the opportunity to comprehensively investigate not only human tumor biology, but also the interactions between the immune system and CAR T cells, and their impacts on adoptive T cell efficacy.

The advantages and disadvantages of in vivo mouse models are summarized in Table 1.

To conclude, since it is challenging to evaluate all the aforementioned parameters in a unique mouse strain, multiple model systems are needed to answer all the unanswered questions related to toxicity, proper CAR T cell and TME interaction, as well as CAR T responses against local and/or systemic disease.

## 4. CAR T Cells for Solid Tumors in the Clinical Setting

In the last decade, the feasibility, safety, and preliminary efficacy of CAR T cells targeting a wide range of tumor antigens pertaining to solid tumors have been evaluated in early-phase trials. However, different from what has been reported for hematologic malignancies, there are several hurdles that currently limit the use of CAR T cells in the treatment of solid malignancies. When developing a CAR construct against neoplastic cells, a first key point concerns the specificity of tumor-associated antigens (TAAs), which should ideally be restricted to malignant cells and should be absent in normal cells, in order to mitigate the risk of on-target off-tumor toxicities. Another issue with TAA is their plasticity, which may lead to antigen loss or mutation, thus providing an antigen escape mechanism. The need for highly specific TAAs recognized by CAR T cells is substantiated by reports of severe toxicities caused by on-target off-tumor CAR T cells. Morgan et al. reported the case of a patient with chemotherapy refractory, metastatic colon cancer enrolled in a phase I/II study evaluating anti-ERBB2 CAR T cells who, 15 min after the CAR T cell infusion, developed symptoms of respiratory failure and died 5 days later [92]. The authors speculated that the event could be related to the recognition by anti-ERBB2 CAR T cells of their target expressed on lung cells. Another phase I/II study tested CAR T cells targeting carbonic anhydrase IX (CAIX) in 12 patients with renal cell carcinoma [93]. In four out of eight treated patients, the concomitant expression of CAIX on the bile duct epithelium (detected by liver biopsies performed after the CAR T cell infusion) led to grade 2–4 hepatic toxicity and, consequently, to treatment discontinuation.

### 4.1. Toxicities

Importantly, CAR T may induce a number of potentially life-threatening side effects, such as CRS and immune effector cell-associated neurotoxicity syndrome (ICANS), hemophagocytosis, and prolonged cytopenias [87]. CRS consists of fever, hypotension, hypoxia, and organ toxicity, which can provoke organ failure in severe cases. ICANS includes several neurological symptoms such as reduced concentration, cognitive disorders, confusion, lethargy, aphasia, agitation, tremor, delirium, seizures, paresis, motor weakness, and/or signs of intracerebral pressure. ICANS commonly arises during or after CRS. A twice a day 10-point neurologic evaluation using the ICE screening tool is recommended for early detection [87]. Of note, a broad consensus statement for physicians with updated comprehensive guidelines for the treatment of toxicities associated with immunotherapies has been recently published [94]. Though the impacts of many factors are still unknown (such as tumor burden, prior treatment, and age), the design of CAR T constructs likely plays a key role in the development of toxicity.

While treatment of these systemic syndromes with steroids and immunosuppressive agents is improving, they nonetheless point out the early developmental stage of this therapeutic approach and the need for improvement.

Moreover, the aforementioned adverse events represent the most common acute toxicities, but given the limited duration of follow-ups to date, delayed or long-term effects of CAR T therapy are still to be defined.

### 4.2. Gastrointestinal Cancers

Gastric cancer represents approximately 9% of all cancers, and is the third cause of cancer deaths worldwide, with a 5–20% overall survival (OS) in advanced stages [95]. Several antigens expressed by gastric cancer cells have been identified: human epidermal growth factor receptor 2 (HER2), mucin 1 (MUC1), carcinoembryonic antigen (CEA), claudin 18.2 (CLDN18.2), and mesothelin (MSLN).

CLDN18.2 can be found in up to 70% of gastric adenocarcinomas, thus emerging as a promising target for CAR T cell therapy. A second-generation, autologous CAR T cell targeting CLDN18.2 was investigated in a first-in-human study on patients with advanced, CLDN18.2-positive gastrointestinal cancers [96], including seven patients with gastric and five with pancreatic metastatic adenocarcinoma. Patients first underwent lymphodepletion with fludarabine and cyclophosphamide, and then received one to five cycles of anti-CLDN18.2 CAR T cells. Neither serious adverse events (AEs) nor treatment-related deaths were reported; treatment-related toxicities included leukopenia and grade 1–2 (CRS). Among 11 evaluable patients, one complete response (CR) and two partial responses (PR) were reported, while five patients achieved a stable disease (SD). The median progression-free survival (PFS) was 133 days. HER2 is a transmembrane glycoprotein that mediates cell proliferation and whose overexpression plays a central role in tumorigenesis [97,98]. A phase I trial published in 2018 by Feng et al. showed the feasibility of delivering anti-HER2 CAR T cells in patients with advanced, HER2-positive (>50% of cells) biliary tract and pancreatic cancer [99]. Eleven patients were treated with one or two doses of anti-HER2 CAR T cells (median dose of CAR+ T cells, 2.1 × 10^6^/kg). AEs were generally mild (grade 1–2), the most common being fever (100%), fatigue (36%), anemia and lymphopenia (27%), and liver function test elevation (18%). Despite the good safety profile, efficacy was limited: only one patient achieved a PR, and five patients achieved an SD; the median PFS was 5 months.

Another potential target for CAR T cells is CEA, a glycoprotein that can be found on epithelial cells of the gastrointestinal tract and lungs that is highly expressed on the surface of cancer cells of the gastrointestinal tract [100]. A phase I, dose-escalating study enrolled 14 patients with various gastrointestinal, metastatic cancers (esophagus, gastric, colorectal, pancreatic cancer) to study feasibility, safety, and early efficacy of a first-generation CAR T cell product targeting CEA. All patients underwent lymphodepletion with fludarabine, with or without cyclophosphamide, and received different doses of CAR T cells followed by intravenous IL-2. Despite some evidence of CAR T cells trafficking to tumor sites, no objective response was observed, the best response being an SD in seven out of 14 patients. After a safety review, the trial was permanently interrupted due to the occurrence, in four patients, of severe pulmonary toxicity, possibly related to the expression of CEA in the lung epithelium, as suggested by immunohistochemistry studies on resected samples of lung tissue. Zhang et al. reported the results of a phase I study investigating escalating doses of systemically delivered anti-CEA CAR T cells in 10 patients with relapsed and/or refractory metastatic colorectal cancer [101]. The evaluated construct proved to be safely deliverable, as no CAR T-cell-related serious AEs were observed, not even on-target off-tumor pulmonary and gastrointestinal toxicities. However, limited efficacy was displayed, as the best response was an SD in seven out of 10 patients.

In 2017, Hege et al. reported the results of two phase I studies conducted between 1997 and 1998 in patients with metastatic colon cancer [102] who received an antitumor-associated glycoprotein-72 (TAG72) CAR T cell product (TAG72 is an oncofetal mucin selectively overexpressed by adenocarcinomas [103]) either intravenously or directly in the hepatic artery at different doses and schedules. A biopsy demonstrating TAG2 expression ≥5% in the cancer tissue was mandatory at enrollment. Despite a significant decrease in TAG72 serum levels (>80%), no radiological evidence of efficacy was observed, with 100% of patients experiencing progressing disease while on trial. The persistence of CAR T+ cells was an issue reported in the trial, with a rapid clearance of CAR T cells coinciding with rising levels of antibodies. Pancreatic cancer is one of the most lethal solid tumors, with a 5-year OS rate around 10% (https://seer.cancer.gov/statfacts/html/pancreas.html, accessed 1st July 2022). The overexpression of the epidermal growth factor receptor (EGFR) is reported in the majority of pancreatic carcinomas (70–90%) and is associated with a worse prognosis.

In a phase I trial, EGFR-targeting CAR T cells were investigated in 16 patients with metastatic pancreatic cancer whose cells were EGFR-positive by immunohistochemistry (>50%) [104]. Patients enrolled were first treated with nab-paclitaxel, subsequently received lymphodepletion with cyclophosphamide, and were then dosed with anti-EGFR CAR T cells (median dose of CAR T cells, 3.48 × 10^6^/kg) for up to three consecutive doses in 6 months. Expectedly, the most frequently reported AEs were mucosal and cutaneous events (63%), pleural effusion (31%), and pulmonary interstitial exudation (19%). Seventy-five percent of patients achieved disease stability, and 25% of them attained a PR; median PFS and OS were 3 and 5 months, respectively. A similar approach, using anti-EGFR CAR T cell therapy preceded by nab-paclitaxel and cyclophosphamide, was adopted in a phase I study enrolling 19 patients with unresectable, advanced, EGFR-positive biliary tract carcinoma [105]. Patients received EGFR-targeting CAR T cells for up to three doses (median number of CAR T+ cells, 2.56 × 10^6^/kg). Pulmonary toxicity occurred in five out of 19 patients, including one grade 4 acute distress respiratory syndrome due to pulmonary oedema accompanied by a marked increase in inflammatory mediators that required the administration of tocilizumab. As far as efficacy is concerned, one patient obtained a CR lasting for 22 months, while the majority of patients (10/19) achieved an SD; median PFS was 4 months.

An ideal CAR T cell target for the treatment of pancreatic ductal adenocarcinomas is mesothelin, a glycoprotein highly expressed by pancreatic ductal adenocarcinomas, ovarian cancers, and malignant pleural mesothelioma, with low levels of expression on mesothelial cells in the peritoneum, pleura, and pericardium [106]. In the phase I study published by Beatty et al., six patients with pancreatic ductal adenocarcinoma received second-generation CAR T cells, which were mRNA-engineered to transiently express anti-mesothelin CAR [107]. The authors reported neither CRS nor neurological toxicity. Unfortunately, no objective response was observed, with the best response being an SD reported in two out of six treated patients.

Liver cancer is the sixth most common cause of cancer, and hepatocellular carcinoma (HCC) is its most frequent subtype. Glypican-3 (GPC3), a member of the heparan sulfate proteoglycan family, is highly expressed in HCC [108]. In two sequential phase I studies, 13 patients with advanced GPC3-positive HCC, determined by immunohistochemistry, received escalating doses of anti-GPC3 CAR T cells after lymphodepletion with cyclophosphamide with or without fludarabine [109]. Despite a rapid CAR+ T cell expansion, persistence was short (median, 19.5 days). CRS was reported in the majority of patients (9/13) and was mostly of grade 1–2, although a fatal (grade 5) event occurred. Out of 13 patients receiving the anti-GPC3 CAR T cell therapy, two patients achieved a PR, and one an SD; the median PFS of patients who obtained a PR was 111 and 99 days, respectively. The median OS of the entire cohort was approximately 12 months.

### 4.3. Genitourinary Cancers and Beyond

Prostate cancer is the second most common cancer in developed countries. Prostate-specific membrane antigen (PSMA) is expressed by almost all prostate cancers, with its expression increasing in poorly differentiated and metastatic cancer cells. Junghans et al. reported the results of a phase I study investigating PSMA-targeting CAR T cells with concurrent infusion of IL-2 in patients with prostate cancer [110]. Six patients were enrolled in the study, and received escalating doses of anti-PSMA CAR T cells: no relevant anti-PSMA toxicities were reported, and preliminary activity was observed in two patients attaining a PR, and in one patient achieving a minimal response.

In the case of ovarian cancer, a phase I study evaluated the use of adoptive immunotherapy employing CAR T cells directed against alpha-folate receptor (FR). The toxicity was mild, but unfortunately the reduction in tumor burden was negligible, most likely because of a limited persistence of the CAR T cells within the first month from infusion [111].

Haas and colleagues reported the results of phase I study using T cells that were engineered with mRNA electroporation for a transient expression of a mesothelin-specific CAR to treat 15 patients with ovarian cancer (n = 5), pancreatic ductal adenocarcinoma (n = 5), or malignant pleural mesothelioma (n = 5) [112]. The majority of the AEs reported with the anti-mesothelin CAR T cell therapy were grade 1–2 fatigue (47%) and nausea (40%); however, four patients developed ascites, and the authors demonstrated the presence of CAR+ DNA in two out of four ascites samples, representing the evidence of a potential on-target off-tumor toxicity. Efficacy was limited, since the best response was SD in 11 out of 15 patients, with a median PFS of 2.1 months. Of note, at 2 months, mesothelin CAR+ T cells were not detectable in nine out of 15 patients.

A different approach to the systemic delivery of CAR T cells relies on the local administration of anti-mesothelin CAR T cells. Adusumilli et al. reported the results of a phase I trial investigating the intrapleural delivery of mesothelin-targeted CAR T cells in patients with pleural cancer from malignant pleural mesothelioma, metastatic lung cancer, or breast cancer [113]. Twenty-seven patients were treated with intrapleural mesothelin-targeted CAR T cells, and 18 of them also received the anti–PD-1 monoclonal antibody pembrolizumab. CRS and neurotoxicity were limited to grade 1–2. Thirteen out of 27 patients achieved at least an SD, with two patients attaining a PR. The median time to next treatment was 15.3 months, and median OS was 17.7 months. In patients who received mesothelin-targeted CAR T cells plus pembrolizumab, the median time to next treatment was not reached, and median OS was 23.9 months.

Despite the availability of several antigens to target solid tumors with CAR T cells, and despite the encouraging preclinical data, the clinical experience in early-phase studies enrolling a small number of patients with various solid malignancies is less promising than that observed in studies in hematologic malignancies. More specifically, data generated so far show limited efficacy of several CAR T constructs, possibly related to impaired T cells trafficking to tumor sites, limited persistence and highly immunosuppressive tumor microenvironment, and potential safety concerns due to on-target off-tumor toxicities. Altogether, these factors challenge the adoption of CAR T cells to treat solid malignancies, and should be addressed in future trials.

### 4.4. Brain Tumors

Trials in most brain malignancies have not been encouraging so far [114]. Glioblastoma (GBM), the most common form of glioma and also the most frequent brain malignancy in adults, carries a poor prognosis due to its robust resistance to conventional chemoradiotherapy [115]. The search for novel therapeutic approaches has included CAR T strategies [116], but efficacy has not yet been demonstrated except in anecdotal instances [117,118]. Reasons for the limited efficacy of CAR T cells in brain malignancies extend beyond the identification of targetable antigens with tumor-specific/enriched expression, and include tumor-cell-intrinsic and tumor microenvironment biological properties, as well as brain-specific immunologic variables. Additionally, systemic side effects hamper the efficacy of CAR T cells against brain tumors, as is observed in most solid tumors (Section 4.1)

While liquid malignancies are often genetically clonal, GBM shows marked intratumoral heterogeneity at the genomic and transcriptional level [119,120,121,122,123]. This property may lead to heterogeneous expression of any antigen targeted by CAR T cells, thus limiting efficacy [124]. The transcriptional heterogeneity can certainly extend to stereotypical mutant proteins that represent neoantigens, and are therefore favorable CAR T targets, such as the mutant epidermal growth factor receptor EGFRvIII [124,125,126,127,128,129], by virtue of their lack of expression in healthy tissues. Furthermore, recent work has demonstrated that overexpression of specific cell surface receptor tyrosine kinases (RTK) with oncogenic properties is encoded by double minutes, whose abundance in cells is plastic and can be adjusted to allow cells to escape targeted therapy [123,130,131,132,133,134,135]. This RTK addiction of GBM and plasticity in expression of different members of the RTK family likely accounts for the failure of any therapy specifically targeting isolated members of the RTK family.

An important biological property of GBM that limits efficacy of immunologic approaches, including CAR T cells, is its highly immunosuppressive microenvironment, characterized by numerous TAMs, MDSCs, and a few lymphocytes [136]. Additionally, Treg cells are abundant and drive CAR T exhaustion, senescence, and anergy [137,138] mediated by immunosuppressive cytokines abundantly present in the GBM environment (TGFb among others) [139,140,141,142,143,144,145,146,147]. Furthermore, this GBM-induced immunosuppression is not just local and confined within the tumor, but rather systemic [138,148]. This immunosuppressive milieu is therefore predicted to act as a hostile environment for CAR T cells, even if delivered to the tumor in sufficient amounts. One key advantage to the CAR paradigm, however, is its modular design, which is evolving to bypass specific immunosuppressive axes as they become apparent in preclinical and clinical studies. For example, CARs have been modified to confer intrinsic resistance to Treg cells [149], and have also been engineered to target the tumor microenvironment (including cancer-associated fibroblasts, tumor vasculature, extracellular matrix, and tumor-associated macrophages [11,14,150,151]) rather than tumor cells themselves, in order to provide an indirect approach to limit tumor progression and promote endogenous antitumor activity. In addition to the immunosuppressive properties of GBM, the fact that the brain is an immune privileged organ, further limits efficacy of immunologic therapies. The paucity of brain lymphatic circulation, which was not demonstrated until very recently, and relative lack of dendritic cells, are just two examples of brain properties that may hinder immunologic therapies [152,153,154,155]. Furthermore, the blood–brain and blood–tumor barriers (BBB and BTB, respectively), may limit the diapedesis and intratumoral localization of systemically administered CAR T cells [116], although preclinical studies have shown that CAR T cells are able to traffic, localize and clonally expand against orthotopic xenografts and syngeneic models of GBM. An additional potential limitation of systemic CAR T cell approaches in GBM arises from the histologic intratumoral heterogeneity of these tumors. While large territories of GBM show microvascular proliferation, other areas are hypoperfused and hypoxic. While GBM tumor cells can metabolically adapt to limiting nutrient and oxygen availability, the limited blood perfusion in such hypoxic niches may deny systemically administered CAR T cells access to these areas, and therefore limit efficacy. The BBB, BTB and spatially heterogeneous vascular density are important biological considerations and potential limitations in systemic therapeutics for GBM. The vast majority of CAR T cell trials in GBM have utilized systemic intravenous administration. However, vis-à-vis these limitations, alternative delivery approaches ought to be considered. As an example, intratumoral and intraventricular administration of CAR T cells targeting either IL13Rα2 in GBM or GD2 in pontine glioma have led to tumor regression in a few patients [7,117,118]. Other trials are employing intraventricular administration of CAR in high-grade glioma targeting EGFRvIII [156] or HER2 [157], with encouraging results. Another delivery approach that bypasses the above limitations is convection-enhanced delivery (CED) [158], in which stereotactically placed catheters in and around the tumor are used to deliver the therapeutic cargo. Indeed, a current trial is evaluating intratumoral CED delivery of CAR T cells targeting EGFRvIII (INTERCEPT trial).

Of note, GBM is not the only brain malignancy that is being tested with CAR T therapies. Other adult and pediatric brain tumors that are being evaluated include other forms of glioma, medulloblastoma, and ependymoma. The most popular targets so far in clinical trials have been EGFRvIII, HER2, IL13Rα2, GD2, B7-H3, chlorotoxin, and EPHA2 (see Table 2 for a list of CAR T trials in glioma/GBM and other primary brain tumors on clinicaltrials.gov). Of note, several trials are now combining the CAR T therapy with immune checkpoint inhibitors targeting PD1 or CTLA4, in an effort to overcome the immunosuppressive properties of GBM and augment the potency of the CAR T cells (Table 2) [159,160,161]. Furthermore, some studies suggested that lymphodepletion prior to CAR T administration may enhance therapeutic efficacy, a strategy that is also being tested in clinical trials [162]. Overall, the initial enthusiasm about CAR T approaches in brain tumors has been tempered by the preliminary data of clinical trials, with only anecdotal reports of successful treatments. It is clear that the approach will have to be improved, refined, and possibly combined with other immunomodulatory agents before it can produce meaningful results. From the biotechnology point of view, the modular design of CARs continues to evolve toward improving immunologic efficacy.

### 4.5. Pediatric Sarcomas

Osteosarcoma (OS) is a rare pediatric primary bone tumor most commonly diagnosed during adolescence due to rapid bone growth. OS is commonly treated with surgery, and chemotherapy [171,172]; however, the development of immunotherapies and adoptive cell transfers, namely chimeric antigen receptor T cells (CAR T), has led to novel targeted therapies for OS [172].

Human epidermal growth factor 2 (HER2) is a widely known tumor antigen commonly associated with breast cancer that is also found in pediatric sarcomas. In vivo studies regarding expression of HER2 in OS is highly debated based on immunohistochemistry and flow cytometry data [173,174]. In addition, its use as a prognostic indicator is also contentious, as some studies suggest it is associated with poor prognosis [174], while others claim a favorable prognosis, or no association at all [175,176]. A review by Gill published in 2021 outlines the data for HER2 expression in OS, and concludes that a subgroup of patients with OS express HER2 [176]. In vitro studies with OS cell lines and HER2 CAR T cells demonstrated the activation of immune responses and death of HER2-positive OS cells lines [157]. The authors of the study in question went on to conduct a phase I/II clinical trial using HER2 CAR T cells to treat 19 relapsed/refractory HER2-positive sarcoma patients, 16 of which had osteosarcoma. The safety and efficacy data published in 2015 show that patients were able to tolerate increasing doses of CAR T cells, with 10 of the 16 OS patients having progression of the disease [177].

Other pediatric sarcoma tumor targets include GD2 and B7-H3. GD2 is a glycolipid found in neuronal stem cells, but is also upregulated in pediatric cancers, including neuroblastoma, Ewing’s sarcoma, rhabdomyosarcoma, and osteosarcoma [178]. GD2 CAR T cells showed therapeutic potential in an in vitro study using an osteosarcoma cell line. Interestingly, OS cell lines exposed to GD2 CAR T cells showed increased cell surface levels of PD-1, suggesting tumor adaptability and immune system escape [179]. In addition, B7-H3, a checkpoint molecule associated with tumor growth, has been identified as a potential CAR T cell target for pediatric solid tumors and sarcomas. In vivo, OS mouse models treated with B7-H3 CAR T cells had increased survival and decreased tumor growth compared to control mice [180].

Overall, clinical outcomes for most pediatric high-risk solid tumors remain very poor, and relapse is dramatically fatal for sarcomas and brain tumors. However, novel immunotherapeutic modalities, such as CAR T cells, hold promise, though clinical benefits remain to be determined [181].

## 5. Conclusions

Following the encouraging results obtained for lymphoproliferative disorders, CAR T cells have become an intensive area of research as a potential curative treatment in solid tumors. However, their real efficacy remains to be determined. A pivotal role is played by the tumor microenvironment, where physical and functional barriers significantly hamper the interactions between cancer cells and immune cells. CAR T cell infiltration into the tumor tissue, or their exhaustion by the immunosuppressive factors typical of the tumor microenvironment, make this form of cell therapy challenging. However, despite the hurdles experienced in both preclinical and clinical studies, one advantage of this cellular therapy is the intrinsic potential to improve its design. Translational research may considerably improve their efficacy and safety in the near future.

## Figures and Tables

**Figure 1 cancers-14-05108-f001:**
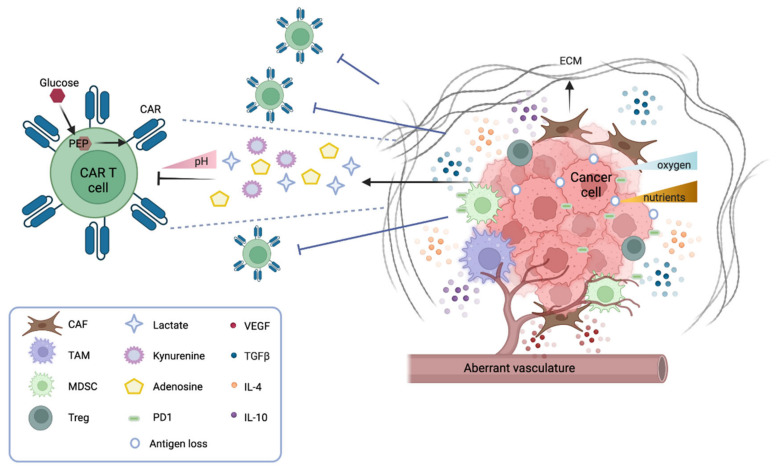
The tumor microenvironment (TME) in solid tumors. Numerous factors in a solid tumor mass inhibit the function of CAR T cells. Cancer cells downregulate their antigens and express immune checkpoint inhibitor molecules (PD1 and others). They also generate an unfavorable microenvironment by secreting metabolites that inhibit the function of CAR T, such as kynurenine, adenosine, and lactate (the latter contributing to lowering the pH). Moreover, tumor cells scavenge nutrients, such as glucose, which, when converted into phosphoenolpyruvate (PEP), are essential for CAR T cell function. Immunosuppressive leucocytes also participate in dampening the function of CAR T cells; tumor-associated macrophages (TAMs), myeloid-derived suppressor cells (MDSCs), and regulatory T cells (T_reg_) secrete inhibitory cytokines (TGFb, IL-4, IL-10) that inhibit CAR T cell activation and favor tumor escape. Another cytokine, VEGF, stimulates the formation of an aberrant vasculature, which contributes to the hypoxic environment hostile to CAR T cell function. The cancer stroma also contains cancer-associated fibroblasts (CAFs) that secrete inhibitory cytokines and the extracellular matrix (ECM), forming a physical barrier to CAR T cell infiltration.

**Table 1 cancers-14-05108-t001:** Benefits and limitations of in vivo model systems.

In Vivo Mouse Models	Advantages	Disadvantages
Syngeneic mouse model		
- Immunocompetent mouse model	On-target and off-tumor toxicity study in mouse system. Orthotopic injection of mouse cancer cell line in corresponding organ (interaction between CAR T cells and TME in mouse system). Non-invasive tumor detection (i.e., BLI).	Lack of tumor heterogeneity. Lack of proper reproducibility of human TME (mouse immune system ≠ human immune system). Limitations for CRS and NE assays. Not suitable to host human tumor cells (human cancer cell lines and PDXs).
Human xenograft mouse model		
- Human-derived cultured tumor cell line xenograft model	Suitable to host human tumor cells. Non-invasive tumor detection (i.e., BLI).	On-target off-tumor toxicity only if CAR construct with double affinity (human tumor antigen and corresponding murine ortholog of mouse tissues). Lack of tumor heterogeneity (human cancer cell lines). Difficulty of reproducibility of human TME in unmodified NSG strain (lack of circulating human immune system). Limitations for CRS and NE assays.
- Patient-derived xenograft mouse model	Tumor heterogeneity. Improved TME reproducibility (orthotopic transplantation; implantation of tumor-associated immune cells and tumor cells from patient).	Obstacles for non-invasive tumor detection (i.e., BLI). Still defective TME: orthotopic transplantation only associated with in situ immunosuppressive stimuli (lack of circulation human immune system). Limitations for CRS and NE assays.
- Humanized xenograft mouse model	Better reproducibility of human TME and interaction between human immune cells and CAR T cells. Proper CRS and NE assays.	High costs. Low, but still present, risk of GvHD onset (shorter mouse life cycle).

For abbreviations see main text.

**Table 2 cancers-14-05108-t002:** List of past and current CAR T trials in primary brain tumors (from clinicaltrials.gov).

Status		Title	Conditions	Interventions	Target	Locations	Trial Number	Relevant Publications	Study Results
**Recruiting**	1	GD2 CAR T Cells in Diffuse Intrinsic Pontine Gliomas(DIPG) & Spinal Diffuse Midline Glioma(DMG)	Glioma of Spinal Cord Glioma of Brainstem	Drug: GD2 CAR T cells | Drug: Fludarabine | Drug: Cyclophosphamide	GD2	Lucile Packard Children’s Hospital (LPCH), Palo Alto, USA	NCT04196413	[118]	Phase 1 trial with 4 patients
	2	Personalized Chimeric Antigen Receptor T Cell Immunotherapy for Patients With Recurrent Malignant Gliomas	Glioma Malignant Glioma of Brain Recurrence Tumor	Biological: chimeric antigen receptor T cells	EPHA2	Xuanwu Hospital, Beijing, China	NCT03423992	[163]	No Results
	5	Study of B7-H3-Specific CAR T Cell Locoregional Immunotherapy for Diffuse Intrinsic Pontine Glioma/Diffuse Midline Glioma and Recurrent or Refractory Pediatric Central Nervous System Tumors	Central Nervous System Tumor Diffuse Intrinsic Pontine Glioma Diffuse Midline Glioma Ependymoma Medulloblastoma Germ Cell Tumor Atypical Teratoid/Rhabdoid Tumor Primitive Neuroectodermal Tumor Choroid Plexus Carcinoma Pineoblastoma Glioma	Biological: SCRI-CARB7H3(s); B7H3-specific chimeric antigen receptor (CAR) T cell	B7-H3	Seattle Children’s Hospital, Seattle, USA	NCT04185038		No Results
	6	Chimeric Antigen Receptor (CAR) T Cells With a Chlorotoxin Tumor-Targeting Domain for the Treatment of MPP2+ Recurrent or Progressive Glioblastoma	Recurrent Glioblastoma Recurrent Malignant Glioma Recurrent WHO Grade II Glioma Recurrent WHO Grade III Glioma	Biological: Chlorotoxin (EQ)-CD28-CD3zeta-CD19t-expressing CAR T-lymphocytes	Chlorotoxin	City of Hope Medical Center, Duarte, USA	NCT04214392		No Results
	7	C7R-GD2.CAR T Cells for Patients With GD2-expressing Brain Tumors (GAIL-B)	Diffuse Intrinsic Pontine Glioma High-Grade Glioma Embryonal Tumor Ependymal Tumor	Genetic: (C7R)-GD2.CAR T cells | Drug: Cyclophosphamide | Drug: Fludarabine	GD2	Texas Children’s Hospital, Houston, USA	NCT04099797		No Results
	8	CD147-CART Cells in Patients With Recurrent Malignant Glioma.	Recurrent Glioblastoma—CD147-Positive	Biological: CD147-CAR T	CD147	National Translational Science Center for Molecular Medicine, Xi’an, China	NCT04045847		No Results
	9	Autologous CAR-T/TCR-T Cell Immunotherapy for Solid Malignancies	Esophagus Cancer Hepatoma Glioma Gastric Cancer	Biological: CAR T/TCR T cell immunotherapy	EGFRvIII	Henan Provincial People’s Hospital, Zhengzhou, China	NCT03941626		No Results
	10	Autologous CAR-T/TCR-T Cell Immunotherapy for Malignancies	B cell Acute Lymphoblastic Leukemia Lymphoma Myeloid Leukemia Multiple Myeloma Hepatoma Gastric Cancer Pancreatic Cancer Mesothelioma Colorectal Cancer Esophagus Cancer Lung Cancer Glioma Melanoma Synovial Sarcoma Ovarian Cancer Renal Carcinoma	Biological: CAR T cell immunotherapy	EGFRvIII	The First Affiliated Hospital of Zhengzhou University, Zhengzhou, China	NCT03638206		No Results
	11	Genetically Modified T-cells in Treating Patients With Recurrent or Refractory Malignant Glioma	Recurrent Glioblastoma Recurrent Malignant Glioma Recurrent WHO Grade II Glioma Recurrent WHO Grade III Glioma Refractory Glioblastoma Refractory Malignant Glioma Refractory WHO Grade II Glioma Refractory WHO Grade III Glioma	Biological: IL13Rα2-specific Hinge-optimized 4-1BB-co-stimulatory CAR/Truncated CD19-expressing Autologous TN/MEM Cells Biological: IL13Rα2-specific Hinge-optimized 41BB-co-stimulatory CAR Truncated CD19-expressing Autologous T Lymphocytes Other: Laboratory Biomarker Analysis Procedure: Magnetic Resonance Imaging Procedure: Magnetic Resonance Spectroscopic Imaging Other: Quality-of-Life Assessment	IL13Rα2	City of Hope Comprehensive Cancer Center, Duarte, USA	NCT02208362	[117]	No Results
	12	EGFR806-specific CAR T Cell Locoregional Immunotherapy for EGFR-positive Recurrent or Refractory Pediatric CNS Tumors	Central Nervous System Tumor Pediatric| Glioma Ependymoma Medulloblastoma Germ Cell Tumor Atypical Teratoid/Rhabdoid Tumor Primitive Neuroectodermal Tumor Choroid Plexus Carcinoma Pineoblastoma	Biological: EGFR806-specific chimeric antigen receptor (CAR) T cell	EGFR806	Seattle Children’s Hospital, Seattle, USA	NCT03638167		No Results
	13	HER2-specific CAR T Cell Locoregional Immunotherapy for HER2-positive Recurrent/Refractory Pediatric CNS Tumors	Central Nervous System Tumor Pediatric| Glioma Ependymoma Medulloblastoma Germ Cell Tumor Atypical Teratoid/Rhabdoid Tumor Primitive Neuroectodermal Tumor Choroid Plexus Carcinoma Pineoblastoma	Biological: HER2-specific chimeric antigen receptor (CAR) T cell	HER2	Seattle Children’s Hospital, Seattle, USA	NCT03500991	[164]	No Results
	14	Memory-Enriched T Cells in Treating Patients With Recurrent or Refractory Grade III-IV Glioma	Glioblastoma Malignant Glioma Recurrent Glioma Refractory Glioma WHO Grade III Glioma	Biological: HER2(EQ)BBζ/CD19t+ T cells | Other: Laboratory Biomarker Analysis | Procedure: Leukapheresis	HER2	City of Hope Medical Center, Duarte, USA	NCT03389230		No Results
	15	Pilot Study of B7-H3 CAR-T in Treating Patients With Recurrent and Refractory Glioblastoma	Recurrent Glioblastoma Refractory Glioblastoma	Drug: B7-H3 CAR T | Drug: Temozolomide	B7-H3	the Second Affiliated Hospital of Zhejiang University School of Medicine, Hangzhou, China	NCT04385173		No Results
	16	NKG2D-based CAR T-cells Immunotherapy for Patient With r/r NKG2DL+ Solid Tumors	Hepatocellular Carcinoma Glioblastoma Medulloblastoma Colon Cancer	Biological: NKG2D-based CAR T cells	NKG2D	Xunyang Changchun Shihua Hospital, Jiujiang, China	NCT05131763		No Results
	17	Brain Tumor-Specific Immune Cells (IL13Rα2-CAR T Cells) for the Treatment of Leptomeningeal Glioblastoma, Ependymoma, or Medulloblastoma	Leptomeningeal Metastases	Biological: IL13Rα2-specific Hinge-optimized 41BB-co-stimulatory CAR Truncated CD19-expressing Autologous T Lymphocytes	IL13Rα2	City of Hope Medical Center, Duarte, USA	NCT04661384		No Results
	18	IL13Rα2-CAR T Cells With or Without Nivolumab and Ipilimumab in Treating Patients With GBM	Recurrent Glioblastoma Refractory Glioblastoma	Biological: IL13Rα2-specific Hinge-optimized 4-1BB-co-stimulatory CAR/Truncated CD19-expressing Autologous TN/MEM Cells Biological: Ipilimumab Biological: Nivolumab Other: Quality-of-Life Assessment Other: Questionnaire Administration	IL13Rα2	City of Hope Medical Center, Duarte, USA	NCT04003649		No Results
	19	B7-H3 CAR-T for Recurrent or Refractory Glioblastoma	Recurrent Glioblastoma Refractory Glioblastoma	Drug: Temozolomide | Biological: B7-H3 CAR T	B7-H3	the Second Affiliated Hospital of Zhejiang University School of Medicine, Hangzhou, China | Huzhou Central Hospital, Huzhou, China | Ningbo Yinzhou People’s Hospital, Ningbo, China	NCT04077866		No Results
	20	Intracranial Injection of NK-92/5.28.z Cells in Patients With Recurrent HER2-positive Glioblastoma	Glioblastoma	Biological: NK-92/5.28.z	HER2	Johann W. Goethe University Hospital, Department of Neurosurgery, Frankfurt, Germany | Johann W. Goethe University Hospital, Senckenberg Institute of Neurooncology, Frankfurt, Germany	NCT03383978		No Results
**Enrolling by invitation**	21	Immunogene-modified T (IgT) Cells Against Glioblastoma Multiforme	Glioblastoma Multiforme of Brain Glioblastoma Multiforme	Biological: Antigen-specific IgT cells	EGFRvIII	Shenzhen Geno-immune Medical Institute, Shenzhen, China | Department of Neurosurgery, Shenzhen Hospital, Southern Medical University, Shenzhen, China	NCT03170141		No Results
**Active, not recruiting**	22	The Efficacy and Safety of Brain-targeting Immune Cells (EGFRvIII-CAR T Cells) in Treating Patients With Leptomeningeal Disease From Glioblastoma. Administering Patients EGFRvIII -CAR T Cells May Help to Recognize and Destroy Brain Tumor Cells in Patients	Glioblastoma Glioblastoma Multiforme Glioma, Malignant	Biological: EGFRvIII-specific hinge-optimized CD3 ζ-stimulatory/41BB-co-stimulatory Chimeric Antigen Receptor autologous T lymphocytes	EGFRvIII	Jyväskylä Central Hospital, Jyväskylä, Finland | University Of Oulu, Oulu, Finland | Apollo Hospital, New Delhi, India	NCT05063682		No Results
	23	HER2-specific Chimeric Antigen Receptor (CAR) T Cells for Children With Ependymoma	Ependymoma	Biological: HER2-Specific CAR T Cell	HER2	Phoenix Children’s Hospital, Phoenix, USA | Children’s Hospital of Los Angeles, Los Angeles, USA | Lucile Packard Children’s Hospital at Stanford University Medical Center, Palo Alto, USA | Children’s Hospital Colorado, Aurora, USA | Children’s National Medical Center, Washington, USA | Children’s Healthcare of Atlanta, Atlanta, USA | Children’s Memorial Hospital, Chicago, Chicago, USA | Lurie Children’s Hospital-Chicago, Chicago, USA | Memorial Sloan Kettering Cancer Center, New York, USA | Cincinnati Children Hospital Medical Center, Cincinnati, USA | Children’s Hospital of Pittsburgh, Pittsburgh, USA | Texas Children’s Cancer Center, Houston, USA	NCT04903080		No Results
**Not yet recruiting**	24	NKG2D CAR-T(KD-025) in the Treatment of Relapsed or Refractory NKG2DL+ Tumors	Solid Tumor Hepatocellular Carcinoma Colorectal Cancer Glioma	Drug: KD-025 CAR T cells	NKG2D	The Affiliated Nanjing Drum Tower Hospital of Nanjing University Medical School, Nanjing, China	NCT04550663		No Results
	25	Pilot Study of NKG2D CAR-T in Treating Patients With Recurrent Glioblastoma	Recurrent Glioblastoma	Biological: NKG2D CAR T	NKG2D	UWELL Biopharma, China	NCT04717999		No Results
	26	CART-EGFR- IL13Rα2 in EGFR Amplified Recurrent GBM	Glioblastoma	Drug: 5 × 10(7) CART-EGFR-IL13Rα2 Drug: 1 × 107 CART-EGFR-IL13Rα2 Drug: 1 × 10(8) CART-EGFR-IL13Rα2 Drug: 5 × 108 CART-EGFR-IL13Rα2 Drug: Cyclophosphamide | Drug: Fludarabine	EGFR/IL13Rα2	Abramson Cancer Center of the University of Pennsylvania, Philadelphia, USA	NCT05168423		No Results
	27	Long-term Follow-up of Subjects Treated With CARv3-TEAM-E T Cells	Glioblastoma Recurrent, EGFR vIII Mutant Newly Diagnosed Glioblastoma, EGFRvIII Mutant Recurrent Glioblastoma, EGFR vIII Negative	Diagnostic Test: Disease assessments Procedure: Tumor Biopsy Diagnostic Test: Blood test	EGFRvIII	Massachusetts General Hospital, Boston, USA	NCT05024175		No Results
**Completed**	28	CART-EGFRvIII + Pembrolizumab in GBM	Glioblastoma	Biological: CART-EGFRvIII T cells Biological: Pembrolizumab	EGFRvIII	Abramson Cancer Center of the University of Pennsylvania, Philadelphia, USA	NCT03726515		No Results
	29	Cellular Adoptive Immunotherapy Using Genetically Modified T-Lymphocytes in Treating Patients With Recurrent or Refractory High-Grade Malignant Glioma	Brain and Central Nervous System Tumors	Biological: therapeutic autologous lymphocytes Genetic: gene expression analysis Other: laboratory biomarker analysis	IL13Rα2	City of Hope Medical Center, Duarte, USA	NCT00730613	[117,165]	Phase I trial
	30	Phase I Study of Cellular Immunotherapy for Recurrent/Refractory Malignant Glioma Using Intratumoral Infusions of GRm13Z40-2, An Allogeneic CD8+ Cytolitic T-Cell Line Genetically Modified to Express the IL 13-Zetakine and HyTK and to be Resistant to Glucocorticoids, in Combination With Interleukin-2	Anaplastic Astrocytoma Anaplastic Ependymoma Anaplastic Meningioma Anaplastic Oligodendroglioma Brain Stem Glioma Ependymoblastoma Giant Cell Glioblastoma Glioblastoma Gliosarcoma Grade III Meningioma Meningeal Hemangiopericytoma Mixed Glioma Pineal Gland Astrocytoma Brain Tumor	Biological: therapeutic allogeneic lymphocytes Biological: aldesleukin Other: laboratory biomarker analysis Procedure: positron emission tomography	IL13Rα2	City of Hope, Duarte, USA	NCT01082926	[166]	
**Terminated**	31	EGFRvIII CAR T Cells for Newly-Diagnosed WHO Grade IV Malignant Glioma	Glioblastoma Gliosarcoma	Biological: EGFRvIII CAR T cells	EGFRvIII	The Preston Robert Tisch Brain Tumor Center at Duke, Durham, USA	NCT02664363	[162]	Phase I trial—3 patients enrolled
	32	Autologous T Cells Redirected to EGFRVIII-With a Chimeric Antigen Receptor in Patients With EGFRVIII+ Glioblastoma	Patients With Residual or Recurrent EGFRvIII+ Glioma	Biological: CAR T-EGFRvIII T cells	EGFRvIII	UCSF, San Francisco, USA | Abramson Cancer Center of the University of Pennsylvania, Philadelphia, USA	NCT02209376	[124,167,168,169,170]	Terminated to pursue combinatorial strategies
	33	Intracerebral EGFR-vIII CAR-T Cells for Recurrent GBM	Recurrent Glioblastoma Recurrent Gliosarcoma	Biological: EGFRvIII-CARs	EGFRvIII	Duke University Medical Center, Durham, USA	NCT03283631		No Results
**Withdrawn**	34	CAR-T Cell Immunotherapy for GD2 Positive Glioma Patients	Glioma of Brain	Biological: GD2 CAR T immunotherapy	GD2	Fuda Cancer Hospital, Guangzhou, China	NCT04406610		No Results
	35	CAR-T Cell Immunotherapy for GD2 Positive Glioma Patients	GD2-Positive Glioma	Biological: CAR T cell immunotherapy	GD2	Central laboratory in Fuda cancer hospital, Guangzhou, China	NCT03252171		No Results
	36	CAR-T Cell Immunotherapy for EphA2 Positive Malignant Glioma Patients	EphA2-Positive Malignant Glioma	Biological: CAR T cell immunotherapy	EPHA2	Central laboratory in Fuda cancer hospital, Guangzhou, China	NCT02575261		No Results
	37	A Clinical Research of CAR T Cells Targeting HER2 Positive Cancer	Breast Cancer Ovarian Cancer Lung Cancer Gastric Cancer Colorectal Cancer Glioma Pancreatic Cancer	Biological: Anti-HER2 CAR T	HER2	Southwest Hospital of Third Military Medical University, Chongqing, China	NCT02713984		No Results
**Unknown status**	38	CAR T Cells in Treating Patients With Malignant Gliomas Overexpressing EGFR	Advanced Glioma	Biological: anti-EGFR CAR T	EGFR	Shanghai Cancer Institute, Xuhui, China	NCT02331693		No Results
	39	CAR-T Cell Immunotherapy in MUC1 Positive Solid Tumor	Malignant Glioma of Brain Colorectal Carcinoma Gastric Carcinoma	Biological: anti-MUC1 CAR T cells	MUC1	PersonGen Biomedicine (Suzhou) Co., Ltd., Suzhou, China	NCT02617134		No Results
	40	CAR-pNK Cell Immunotherapy in MUC1 Positive Relapsed or Refractory Solid Tumor	Hepatocellular Carcinoma Non-small Cell Lung Cancer Pancreatic Carcinoma Triple-Negative Invasive Breast Carcinoma Malignant Glioma of Brain Colorectal Carcinoma Gastric Carcinoma	Biological: anti-MUC1 CAR-pNK cells	MUC1	PersonGen BioTherapeutics (Suzhou) Co., Ltd., Suzhou, China	NCT02839954		No Results
	41	Pilot Study of Autologous Anti-EGFRvIII CAR T Cells in Recurrent Glioblastoma Multiforme	Glioblastoma Multiforme	Biological: anti-EGFRvIII CAR T cells | Drug: cyclophosphamide | Drug: Fludarabine	EGFRvIII	Sanbo Brain Hospital Capital Medical University, Beijing, China	NCT02844062		No Results
	42	Pilot Study of Autologous Chimeric Switch Receptor Modified T Cells in Recurrent Glioblastoma Multiforme	Glioblastoma Multiforme	Biological: Anti-PD-L1 CSR T cells | Drug: Cyclophosphamide | Drug: Fludarabine	PD-L1	Sanbo Brain Hospital Capital Medical University, Beijing, China	NCT02937844		No Results

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
