# Peer review of "Advances and Hurdles in CAR T Cell Immune Therapy for Solid Tumors"

_cancers, 2022, doi:10.3390/cancers14205108_

Round 1

Reviewer 1 Report

Brief Summary

This review discusses key areas of interest related to CAR T cell therapy for solid tumors, particularly highlighting the challenges of overcoming an immune suppressive microenvironment and optimizing models for preclinical in vivo analysis to enhance translation.  Results of preliminary clinically studies for CAR T cells in Gi cancers, GU cancers, brain tumors, and pediatric solid tumors are discussed.

General Concept Comments

The overall scope of the review is relevant, highlighting key challenges to the application of CAR T cells in solid tumors and identifying strategies for improvement currently undergoing clinical evaluation. References are current and support the authors’ conclusions.

In the discussion of in vivo modeling, recommend acknowledging that there is currently not one ideal model that in isolation can adequately address modeling impact of the tumor microenvironment, evaluating efficacy against local and systemic disease, and evaluating for on-target off-tumor toxicity, while directly assessing a potential construct for clinical use. Thus, potential CAR constructs and therapeutic strategies will likely require evaluation in multiple model systems to allow for safe and effective translation to early phase clinical studies.

Within the Clinical Evaluations section, the format and approach to the section on brain tumors is not consistent with the other sections, would improve flow if these were parallel. Additionally, there is a section labeled as “Osteosarcoma,” but in general the approach for clinical studies is antigen rather than tumor-type specific. In example the referenced studies treated a range of pediatric sarcomas. In addition, GD2 targeting is being clinically evaluated in neuroblastoma. It would be more representative to entitle this section “Pediatric Solid Tumors” and discuss the strategies broadly.  

Specific Comments:

2. Solid Tumor Microenvironment

Figure 1 (line 54)

            -define phosphoenolpyruvate (PEP) in figure legend

2.1 (line 73)

            -include reference demonstrating tumors down-regulating chemo-attractant molecules

2.2 (line 119)

-components of CAR design including choice of costimulatory molecule can impact reliance on glycolytic vs. aerobic metabolic pathways (Kawalekar et al, Immunity 2016, PMID 28843072)

3. In vivo Models

-consider adding summary figure identifying benefits and limitations of in vivo model systems

3.1 (line 231-232)

-would note that a limitation of syngeneic models for safety evaluation is that the murine construct being evaluated may be based on but not identical to the construct being used in future clinical application

4. Clinical Evaluations

4.1 (line 397)

-would comment that these are the most common acute toxicities, but given limited duration of follow-up to date delayed or late effects of CAR T cell therapy are not well defined

4.3 consider additional GU reference:

-folate receptor alpha for ovarian cancer (Kershaw, et al, Clin Cancer Res 2006, PMID 17062687)

4.4 (lines 540-547)

-This discussion of systemic side effects of CAR T therapy is repetitious of section 4.1 and not specific to brain tumor population, would remove from this section.

4.4 (lines 559-570)

-Much of the general discussion of impact of immune suppressive environment overlaps with section 2, in this section would focus on the components specific to brain tumors.

4.4 (line 595)

            -Add references to specific brain tumor CAR trials evaluating IV administration:

HER2 for high grade glioma (Ahmed et al, JAMA Oncol 2017, PMID 28426845)

EGFRvIII for high grade glioma (Goff et al, J Immunother 2019, PMID 30882547)

4.4 (lines 611-613)

-Role of lymphodepletion in CAR T cell therapy has been established more broadly, this statement could be moved to the section overview as it applies beyond brain tumors.

4.4 (line 607, 610)

            -Table 1 is referenced, but is not included in the available manuscript documents

4.5 (lines 620-630)

-as above, would remove this detailed background of osteosarcoma and instead include a more general reference to outcomes for relapsed/refractory pediatric solid tumors.

Author Response

Reviewer 1 Comments and Suggestions for Authors

Brief Summary

This review discusses key areas of interest related to CAR T cell therapy for solid tumors, particularly highlighting the challenges of overcoming an immune suppressive microenvironment and optimizing models for preclinical in vivo analysis to enhance translation. Results of preliminary clinically studies for CAR T cells in Gi cancers, GU cancers, brain tumors, and pediatric solid tumors are discussed.

General Concept Comments

The overall scope of the review is relevant, highlighting key challenges to the application of CAR T cells in solid tumors and identifying strategies for improvement currently undergoing clinical evaluation. References are current and support the authors’ conclusions.

We thank Reviewer 1 for their comments and suggestions, which we tried to address as detailed. In the discussion of in vivo modeling, recommend acknowledging that there is currently not one ideal model that in isolation can adequately address modeling impact of the tumor microenvironment, evaluating efficacy against local and systemic disease, and evaluating for on- target off-tumor toxicity, while directly assessing a potential construct for clinical use. Thus, potential CAR constructs and therapeutic strategies will likely require evaluation in multiple model systems to allow for safe and effective translation to early phase clinical studies.

We fully agree with Reviewer 1 observation, and therefore we added a paragraph at the end of section 3.1 (rows 312-317) detailing the limitations of current approaches.

Within the Clinical Evaluations section, the format and approach to the section on brain tumors is not consistent with the other sections, would improve flow if these were parallel. Additionally, there is a section labeled as “Osteosarcoma,” but in general the approach for clinical studies is antigen rather than tumor-type specific. In example the referenced studies treated a range of pediatric sarcomas. In addition, GD2 targeting is being clinically evaluated in neuroblastoma. It would be more representative to entitle this section “Pediatric Solid Tumors” and discuss the strategies broadly.

We agree with Reviewer 1 comment, and indeed the section is broader than the initial title suggested. This section has therefore been renamed “Pediatric Sarcomas” to reflect the larger variety of tumors that can be targeted by CAR T treatment. We also included a closing statement (and additional reference #181) on the promising use of CAR T for pediatric sarcomas.

Specific Comments:

  1. Solid Tumor Microenvironment

Figure 1 (line 54)

-define phosphoenolpyruvate (PEP) in figure legend

We thank the reviewer for noticing, the definition has been added

2.1 (line 73)

-include reference demonstrating tumors down-regulating chemo-attractant molecules

The following reference has been added

Beatty, G. L. & Gladney, W. L. Immune escape mechanisms as a guide for cancer immunotherapy Clin

Cancer Res 21, 687-692, doi:10.1158/1078-0432.CCR-14-1860 (2015)

2.2 (line 119)

-components of CAR design including choice of costimulatory molecule can impact reliance on glycolytic vs. aerobic metabolic pathways (Kawalekar et al, Immunity 2016, PMID 28843072)

We thank the reviewer for this valid point. The observation and reference have been added according to the reviewer’s suggestion.

  1. In vivo Models

-consider adding summary figure identifying benefits and limitations of in vivo model systems

We took into consideration this valid suggestion and therefore we added a table at the end of section 3 to summarize the advantage and disadvantages of specific in vivo models (table 1).

3.1 (line 231-232)

-would note that a limitation of syngeneic models for safety evaluation is that the murine construct being evaluated may be based on but not identical to the construct being used in future clinical application

We appreciate the observation from Reviewer 1, and therefore we added a paragraph (rows 312- 317) to comment on the differences between the murine and human constructs and their future use in the clinical setting.

  1. Clinical Evaluations

4.1 (line 397)

-would comment that these are the most common acute toxicities, but given limited duration of follow-up to date delayed or late effects of CAR T cell therapy are not well defined

We thank the reviewer for this insight. A statement according to the suggestion has been added (rows 466-468).

4.3 consider additional GU reference:

-folate receptor alpha for ovarian cancer (Kershaw, et al, Clin Cancer Res 2006, PMID 17062687)

We thank the reviewer for the observation. The suggested study was added and cited.

4.4 (lines 540-547)

-This discussion of systemic side effects of CAR T therapy is repetitious of section 4.1 and not specific to brain tumor population, would remove from this section.

We agree with the reviewer’s observation. This observation was removed and consolidated within

chapter 4.1.

4.4 (lines 559-570)

-Much of the general discussion of impact of immune suppressive environment overlaps with section 2, in this section would focus on the components specific to brain tumors.

According to the reviewer’s suggestions, this section has been restructured and it now focuses specifically on the GBM microenvironment.

4.4 (line 595)

-Add references to specific brain tumor CAR trials evaluating IV administration:

HER2 for high grade glioma (Ahmed et al, JAMA Oncol 2017, PMID 28426845)

EGFRvIII for high grade glioma (Goff et al, J Immunother 2019, PMID 30882547)

We thank Reviewer 1 for their suggestion. Accordingly, we referenced these additional trials.

4.4 (lines 611-613)

-Role of lymphodepletion in CAR T cell therapy has been established more broadly, this

statement could be moved to the section overview as it applies beyond brain tumors.

Following the reviewer’s observation, we added a statement on lymphodepletion in the general section regarding the solid TME (rows 250-252). On the other hand, we decided to maintain the old citation in place, since it specifically refers to brain tumors.

4.4 (line 607, 610)

-Table 1 is referenced, but is not included in the available manuscript documents

We thank Reviewer 1 for noticing. Table 1 (now Table 2) has been integrated in the final manuscript file.

4.5 (lines 620-630)

-as above, would remove this detailed background of osteosarcoma and instead include a more general reference to outcomes for relapsed/refractory pediatric solid tumors.

In agreement with the reviewer’s suggestion, we restructured the background part of this section and renamed it “Pediatric sarcomas”.

Reviewer 2 Report

The article is an academic work. It would be better if it has more recent references.

Author Response

We thank the reviewer for their comment and suggestion. Ten new references have been added including recent studies (published within the last five years) to better reflect the advances in the field.

Reviewer 3 Report

1.       Current manuscript is too general. It should be developed more extensively.

2.       Section “CAR T Cells for Solid Tumors in the Clinical Setting”: this section is too long and should be more precise and concise. It would suggest the data of this section be summarized in a table.

3.       The title of the manuscript offers advances and hurdles of Car T cell therapy in solid tumors. The hurdles and advances (or overcoming strategies) are lacking. Only toxicity and TME have been discussed.

Author Response

Reviewer 3. Comments and Suggestions for Authors
Current manuscript is too general. It should be developed more extensively.
We thank Reviewer 3 for their observation. The manuscript has been extensively revised to accommodate new recent studies and findings. Furthermore, we added ten recent references to reflect the advances in the field.
2. Section “CAR T Cells for Solid Tumors in the Clinical Setting”: this section is too long and should be more precise and concise. It would suggest the data of this section be summarized in a table.
We thank Reviewer 3 for their comment. We consider this section needs to be longer to reflect the abundance of clinical studies that have lead to meaningful advances in the field. Nevertheless, a comprehensive table (Table 2) has been generated summarizing clinical trials that have been generated to target primary brain
tumors.
3. The title of the manuscript offers advances and hurdles of Car T cell therapy in solid tumors. The hurdles and advances (or overcoming strategies) are lacking. Only toxicity and TME have been discussed.
According to a previous discussion with the guest editor, we chose the current title according to the proposed theme for this special issue. We believe the title of the manuscript actually reflects its content, since the hurdles and advances are represented not only in section 2 (TME) and section 4.1 (toxicities in the clinical setting) but also by the studies conducted in vivo on numerous models (section 3) as well as in the clinical setting (section 4), where they are integrated within the description of each target or tumor type. The manuscript has undergone extensive editing and we believe that hurdles and advances are objectively represented in the current version.

Round 2

Reviewer 3 Report

No comment.